# Migrant mothers' experiences of postnatal depression in the UK

**Laetitia Rater**[1], **Debra Marais**[1*], **Aisling Kelly**[2]

**1** Doctorate of Clinical Psychology Programme, School of Health, Medicine and Life Sciences, University of Hertfordshire, Hatfield, United Kingdom, **2** Maternity Trauma and Loss Care Service, Central and North-West London NHS Foundation Trust, London, United Kingdom

* d.marais@herts.ac.uk

## Abstract

### Background

Despite the majority of migrants coming to the UK for voluntary reasons such as study, work or family, few studies explore their experiences of mental health. Prevalence of postnatal depression (PND) is higher amongst migrant women compared with non-migrant women. This qualitative study aimed to explore voluntary migrant mothers' experiences of postnatal depression (PND) in the UK.

### Methodology

Seven migrant mothers who had experienced PND participated in individual semi-structured interviews. These were analysed using an Interpretative Phenomenological Analysis (IPA) approach which enabled an in-depth exploration of underlying meanings and significance of participant experiences.

### Findings

Four Group Experiential Themes were identified along with associated subthemes. These captured the layers of disconnect, isolation, despair and grief participants reflected on when sharing their experiences of PND. The stories shared were embedded in participants' intersecting identities of being both mothers and migrants. The themes also captured testaments of resilience and growth through an ongoing journey of healing.

### Discussion

This research builds on existing literature looking at migrant women's experiences of PND, by focussing on voluntary migrant mothers in the UK. The findings suggest that migrant mothers who experience PND would particularly benefit from community support in addition to tailored clinical interventions. Further research and clinical

**Data availability statement:** Data cannot be shared publicly because ethics approval was not granted for this. Participants did not provide consent to share their data. Given the identifiability risks inherent with qualitative data and the sensitive nature of this topic, it was not deemed appropriate to publicly share raw data. Data requests may be sent to the following: Ethics Committee with Delegated Authority (ECDA): Health, Science, Engineering and Technology ECDA: ethicsadmin@herts.ac.uk.

**Funding:** The author(s) received no specific funding for this work.

**Competing interests:** The authors have declared that no competing interests exist.

implications are discussed to help improve migrant mothers' experiences of seeking help for PND in the UK.

## Introduction

Migrants make up approximately 16% of the United Kingdom's (UK) total population [1]. Despite the majority of migrants being considered as 'voluntary' – approximately 87%, compared with those considered as 'forced'- few migrant health studies report on their experiences [1,2].

While it is beyond the scope of this paper to discuss debates around terminology with respect to migration, we have referred to voluntary migrants in this paper as those "who change their country of usual residence […] where the decision to migrate is taken freely by the individual concerned, for reasons of 'personal convenience' and without intervention of an external compelling factor" [3]. This definition there-fore does not refer to "refugees, displaced or others forced or compelled to leave their homes" [3]. Although the term 'migrant' can be evocative at times within certain social discourses, it was chosen as the most consistent and clear term to use for the purposes of this research.

Migration, whether voluntary or involuntary, is considered a significant life tran-sition, which creates uncertainty and can negatively influence mental health [4,5]. Whilst both migrant men and women are susceptible to developing psychological difficulties, migrant women seem to be disproportionately affected [6]. Mothers are particularly vulnerable to these stressors which may lead them to develop mental health difficulties in the postpartum period [7]. They are also considered dispropor-tionately impacted compared to fathers, with them being more likely to be the primary caregiver [8]. Becoming a mother "creates new needs, challenges and responsi-bilities which can be particularly difficult to meet when marginalised economically, socially and legally" [9].

Migrant mothers are faced with particular challenges and higher risk with regards to their overall physical and mental health, from pregnancy to delivery and throughout the postnatal period [10]. There is increased risk of complications during pregnancy and birth, as well as a nearly 20% higher prevalence for Postnatal Depression (PND) in migrant mothers compared with native-born mothers [7,10,11].

Multiple theoretical positions have been suggested when exploring the nature and experiences of PND [12]. From a medical perspective, PND is classified as an 'illness' or 'disorder' [13] within the medical model [14,15]. It is defined as "a sus-tained depressive disorder in women following childbirth" [16], characterised by persistent low mood, loss of interest, fatigue, sleep and appetite changes, irritability, poor concentration, guilt, worthlessness, and anxiety [14]. While PND is recognised within diagnostic frameworks, [14,17] diagnoses can at times oversimplify complex experiences by focussing on symptomology rather than the meaning attributed to the distress experienced.

Psychosocial theories instead emphasise stressors and interpersonal challenges as central to the development, experience and impact of PND [12]. With this in mind,

as migrant mothers are particularly vulnerable to facing psychosocial risk factors such as migration-related mental health issues, poor relationships, low socioeconomic status, and limited support, they may be at a high risk of developing PND [18–20]. Although neither the biological nor the psychosocial approaches are sufficient alone to fully explain the nature of PND and to understand women's varied and complex experiences, the psychosocial theoretical model of understanding is most in line with the current study's epistemological critical realist lens to exploring migrant mothers' experiences of PND [21–23].

Experiencing postnatal depression can be challenging in itself for any mother; but for migrant mothers, the layers of adjustment and additional migration-related barriers pose further risks and challenges impacting overall mental health [23,24]. This study explored the lived experiences of migrant mothers with PND in the UK, focusing on the interplay between migration-related factors and perinatal mental health. It aimed to identify both the specific challenges faced and the resilience strategies employed, addressing a gap in the literature by centering in-depth accounts from this underrepresented group.

## Methodology

### Design

An Interpretative Phenomenological Analysis (IPA) approach was chosen for this study due to its idiographic nature, meaning there is a focus on the individual's unique experiences rather than an aim to construct generalised patterns or claims about a phenomenon [25]. Using IPA allowed for attending to the interpretative nature of experiences as well as the external reality which influences those interpretations [26]. IPA not only acknowledges subjectivity, but it values its inherent nature and recognises its utility [27].

This method is in line with a critical realist epistemological stance, meaning that, throughout this research there has been an acknowledgement of experiences, events and causal mechanisms which shaped and influenced the interpretations and overall findings [26]. Critical realism combines an ontological realism, which is concerned with the existence of an external reality outside of human interpretation, and epistemological relativism, which suggests that knowledge is socially constructed through human perspectives and context [28]. In relation to this study, we adopted the position that there is a reality of PND which exists independently to individual experiences whilst also challenging the notion that there is only one truth to this reality [29]. In line with a critical realist epistemology, using IPA allowed for attending to the interpretative nature of experiences as well as the external reality which influences those interpretations [26].

Furthermore, PND can be conceptualised along a continuum of distress [30,31]. As such, this study focused on exploring women's subjective experiences of PND, rather than PND as a dichotomous disorder assessed against diagnostic criteria.

### Positionality

I led this research as part of my doctoral degree and was guided by my co-author supervisors, who held their own distinct positionalities, which were discussed regularly within supervision. As an 'insider-outsider' first author and primary researcher, I could relate to certain attributes and characteristics of participant identities and experiences, whilst also feeling foreign to other aspects [32,33].

I relate to being an 'insider', because of the migrant part of my identity, which has enabled fostering connections and familiarity to migrant communities since a young age [34]. However, migrants are a diverse and heterogenous group. My positionality is also influenced by aspects of my identities such as being a white, middle class, cis gendered female, and raised in only Western societies. I recognise that the intersections of my privileges shaped my experience of migration in a way which may differ from participants in my study and the wider range of migrant communities.

Some key differences between myself and some of the participants are that I first migrated at an early age, providing me with the opportunity to become bilingual in English, learn to assimilate with the Anglophone culture early on, and benefit from accessing well-resourced support systems. If I had not intentionally reflected on these privileges, they may have precluded my ability to understand the unique challenges that participants faced when moving to the UK as adults and/or underestimated the impact that they felt the lack of support and belonging had on their experiences of postnatal depression. The experience of PND is also one that I cannot relate to, as I am not a mother, reinforcing my 'outsider' position.

At times, my pre-existing beliefs around what it is like to be a migrant and parented by a migrant mother risked assuming understanding of my participants' experiences, prompting emotional involvement which could have led to a loss of curiosity and consistency within the interview process [35]. To mitigate assumptions which may have stemmed from my own experiences as a migrant, I regularly engaged in reflexivity [36].

## Reflexivity

Reflexivity is not only important, but "fundamentally intertwined with qualitative research processes" [37]. To be a reflexive researcher is to have the ability to consciously and deliberately look inwards and self-examine with the aim to foster a continual awareness of oneself within the wider context of the research being conducted [36,38].

The skill of reflexivity is improved on through the explicit repeated process of identifying one's beliefs, cognitive biases and emotional responses, as well as the impact this has on research processes and how this can be in turn addressed [36]. In this study, this included multifaceted practices of self and team reflexivity, whereby reflections were shared through bracketing in a reflective journal and throughout supervision discussions with my two co-author supervisors, both of whom held and reflected on their own insider/outsider positionality in relation to the research [35,37].

I noticed that my interpretations, analysis and choice of themes were influenced by my critical realist epistemological stance and my beliefs, which were shaped by my own experience of migration. I was aware of the 'reality' of the current political climate impacting immigration and mental health experiences of care, as well as the evolving narratives and societal expectations around motherhood. I noticed that, at times, my biases led me astray throughout the iterative process of my analysis, as they influenced me to place greater importance on the participants' migration journey and context compared with their specific experience of PND as migrant mothers. Through the practice of reflexivity, I was able to bracket these biases and reframe my thinking by recentring my research question at the core of my interpretations. I used visual aids to help with this, for example by centring the research question within each step of generating themes. This enabled me to compile the findings using a primary lens of migrant mothers' experiences of PND rather than PND experiences which happen to be told by migrant mothers.

## Ethical considerations

Ethics approval for this study was granted by the University of Hertfordshire Ethics Committees with Delegated Authority (ECDAS): Health, Science, Engineering and Technology [aLMS PGR UH 05737(1)]. The risks, consent and confidentiality terms were outlined within the participant information sheet, consent and debrief forms. Participants were invited to a pre-screening call to discuss any queries or concerns, following which those who wanted to proceed provided written consent and interviews were scheduled. Participants were asked to find a quiet, safe and confidential space for the online interview and a distress protocol was made available.

Data was collected and stored in line with the Data Protection Act 2018 and the General Data Protection Regulation (GDPR) 2016, on a secure online drive. All interview transcripts were anonymised by using pseudonyms and redacting any identifiable information.

## Participants

In line with the project timeline, a mixed purposive and snowballing sampling method was used to recruit participants via social media platforms between 25/07/2024 and 09/01/2025 (S1 File). A research advertisement poster was circulated via

my professional social media accounts (e.g., LinkedIn, X, Instagram) which included eligibility criteria. Administrator consent was sought and granted to post on targeted group pages via Facebook and Whatsapp. Prospective participants were able to contact me directly and arrange a screening call via a booking platform (S2 File).

Seven migrant mothers were recruited for this study, which was deemed an appropriate number of participants (between 6–10) for an IPA study [25]. The final sample size of 7 was reached following several rounds of recruitment and with no further participants coming forward, thereby balancing methodological guidance with pragmatic considerations in the time constraints of the Doctorate.

Participants met the following criteria: identifying as mothers, aged 18 or above, fluent in English, having voluntarily migrated to the UK as adults, and have previously self-identified or been diagnosed with PND. Their countries of origin ranged across Asia (1), Europe (2), South America (2) and North America (2). Participants were aged between 34 and 45 years old. They reported having experienced PND within the last 10 years and there was a mixture of participants self-identifying this versus receiving a diagnosis.

### Data collection procedure

In-depth interviews were the chosen data collection method because they provided the intimate focus required to elicit stories around PND experiences of migrant mothers [25]. The interviews were held online as this allowed for flexibility around participants' needs for childcare as well as mitigating geographical constraints [39]. Interviews were recorded with participant verbal and written consent.

The interview guide included a brief introduction and seven open-ended questions, which were divided into three overarching themes: the experience of moving and adapting to the UK, the experience of PND, and the intersection of experiencing PND as a migrant in this country (S3 File).

On average, interviews lasted just over an hour, with the shortest lasting 41 minutes and the longest 78 minutes. At the end of the interview, a debrief form was sent to participants along with a voucher as remuneration for their time.

### Analysis

The recorded interviews were transcribed via MS Teams using intelligent speech recognition technology. A detailed manual checking and re-reading of the transcripts was completed [40]. The data was then analysed manually using Microsoft Word. This facilitated deep immersion in the data, which is central to qualitative research and a crucial element of IPA. The seven steps of IPA included: reading and re-reading the transcript, exploratory noting, constructing experiential statements, searching for connections across those statements, naming and organising the Personal Experiential Themes (PETs), repeating this process with each transcript and finally developing Group Experiential Themes (GETs) from the PETs across all transcripts [25].

### Findings

Four Group Experiential Themes (GETs) were generated from the analysis and subcategorised into subthemes to capture migrant mothers' individual and collective experiences of PND in the UK. Table 1 below summarises these GETs and subthemes.

### GET 1: Navigating layers of disconnect

This GET illustrates the multiple layers of disconnect experienced by participants, which exacerbated their sense of isolation and loneliness throughout their journey of PND. The subthemes differentiate these layers of disconnect whilst also remaining interlinked with an interwoven sense of recurrent loneliness.

**Table 1. Summary of GETs and Subthemes.**

| GETs | Subthemes |
|---|---|
| **GET 1: Navigating layers of disconnect** | • "It doesn't feel like <u>my</u> culture": Cultural displacement<br>• Grappling with old and new relationships<br>• Feeling estranged<br>• "Always around but not present": Loss of self |
| **GET 2: The burden of "learning the ropes by yourself"** | • "Building everything, from nothing"<br>• An isolating silence<br>• Longing for a 'village'<br>• "It's just all on me": Bearing the weight of sole responsibility |
| **GET 3: Unrelenting confrontation: "Another wave was pushing me down"** | • "Everything's survival"<br>• Spiralling into despair<br>• Grieving expectations |
| **GET 4: A winding journey of resilience** | • Finding strength in solidarity<br>• A liberating shift towards healing<br>• "I'm still putting myself together": the struggle continues |

A sense of disconnect appeared to be at the core of these experiences of PND, through the intersections of *migrant* and *mother* identities. This left participants feeling increasingly isolated from others around them and their environment during their period of PND.

**"It doesn't feel like my culture": Cultural displacement.** Participants spoke to an overall sense of displacement and cultural alienation, which seemed to have originated within the migration experience. Lana, who learned English before migrating to the UK, continued to feel 'othered' through language differences, amongst other socio-cultural elements, and seemed adamant that this sentiment would be unlikely to subside over time, setting her apart from British peers:

"I feel like I will never 100% understand the British culture because I wasn't born here, I wasn't raised here. […] I will never speak like a British person or will never have a British accent or I will never fully understand the culture".

The sense of alienation prevailed throughout motherhood, where differences between their home culture and British culture became more evident and impactful.

"You're isolated quite a lot and I think motherhood in this country is very different… from back home […] In [country of origin], if you have a baby then your aunts […], cousins […], friend pop by, you feel very comforted by how well sur-rounded you are versus in London, where it felt really, really lonely." (Valeria).

**Grappling with old and new relationships.** At the same time as navigating cultural displacement, participants referred to the physical and emotional distance from their roots and previously established relationships. Most participants struggled to deal with the relational disconnect which was created, at a time when they most needed to feel understood and heard.

"The distance […] feels like it's multiplied by 100 since having the kids". *(Rosie).*

For some, it may have been these relational losses which motivated them to seek new, local connections. However, what may have initially been perceived as a hopeful endeavour, appeared more difficult in reality, making the experience of motherhood even more challenging:

"It's very difficult to make those friendships and the network, the support, or the <u>village</u> around you. […] It takes a <u>flip-ping</u> long time to build any kind of relationship" (Valeria).

"In my attempts to connect with people here, nobody's connected back enough to where we felt comfortable asking for childcare." (Monica).

**Feeling estranged.** Beyond the struggles to connect with the local community, participants appeared to find themselves disconnected even from those closest to them. This may have been particularly challenging as their support system was already very limited, potentially adding pressures within the home and straining nuclear family relationships.

"We (my husband and I) were like two strangers in the same house". (Lana).

For Sofi, the depths of loneliness that she experienced were not necessarily about being physically alone, but more to do with an overall emotional detachment:

"I wasn't alone. My husband was here with me. My baby was here, […] but the <u>loneliness</u>, of the <u>mothering</u> and the <u>motherhood</u> experience was […] just numbing and paralysing."

**"Always around but not present": Loss of self.** The sense of displacement and disconnect seemed to have become more apparent throughout participants' experiences of motherhood and PND, along with a realisation that this was taking its toll on their sense of self. Becoming a mother was experienced as "such a big… <u>shift</u> […] in identity" (Melany) for some Navigating the new role of becoming a mother appeared to be challenging for most. Some felt that a new self, as a mother, emerged and completely took over their old sense of self:

"I definitely lost my own identity… when having my kids. I'm not Monica anymore. I'm mummy" (Monica).

Others simply referred to a disconnection from the self entirely, leading to a state of near dissociation within the context of PND:

"It's like […] you're not connected to your instinct anymore. You're becoming disconnected from your mother instinct" (Aisha).

## GET 2: The burden of "learning the ropes by yourself"

This GET speaks to the experiences of participants who voiced the challenges they faced around having to "learn the ropes" (Aisha) of becoming migrant mothers by themselves and the unique pressures they endured in doing so whilst also navigating PND.

**"Building everything, from nothing".** With similar struggles to Monica who had to "make this house a home", Valeria found it challenging to do this entirely from scratch in a new country:

"Building from scratch is learning everything […] from scratch. […] It's like it's a blank page and building everything, from nothing."

Learning to navigate a new country, away from the usual support system, whilst also learning to become a new mother seemed to add multiple layers of overwhelm:

"I think the first weeks my son was born because it was so overwhelming, I was tired and just doing everything on my own, I didn't notice but then when days started being the same and my husband was at work and I was… feeling actually I don't have anyone around." (Lana).

This suggests the confronting loneliness participants were faced with whilst juggling the relentless responsibilities and pressures of becoming a mother as a migrant.

"You're basically on your own <u>learning the ropes by yourself</u>… and especially for something that actually requires so much support around you" (Aisha).

Participants expressed feeling lost and disillusioned with the lack of direction they were given in navigating new motherhood.

"I don't feel that… I was… <u>prepared</u> or <u>informed</u>… about… what support I can get as a <u>mother</u>". (Sofi).

This absence of support and guidance during the crucial early months of motherhood, seemed to have a significant impact on participants' moods, with Lana remembering: "I just burst into tears […] because of the way I feel and I can't even speak about it with anyone because I don't have anyone to talk to".

**An isolating silence.** Participants reflected on the impact that silence and stigma around PND had on them. "I felt alone with my struggles, or I thought that it (PND) was […] a rarity. […] I didn't see my experience reflected back to me, in my community" (Sofi).

For some, the lack of opportunities to feel validated and understood left them feeling alone in their experience:

"A lot of people don't realise that even though it's normal, i.e., a lot of people feel that way. It doesn't <u>have to be</u> that way and that that actually is depression rather than just like, oh, 'it's tough being a mom'" (Melany).

Without this external validation in recognising PND or hearing about similar experiences, participants appeared to internalise their struggles in isolation. The isolating silence seemed to perpetuate a sense of rumination and self-blame:

"It's me, I'm doing something wrong, I'm a terrible mother" and "I suck at this, I've sucked from the beginning, I'm terrible" (Monica).

"[…] I just brushed it off, that I just couldn't <u>manage</u>, or I just couldn't <u>live up</u> to the standard or the norm. So I just brushed it off with "I'm just not good enough" (Sofi).

**Longing for a 'village'.** Most participants' sense of increased isolation throughout their PND experiences was also intrinsically linked to being physically far away from their families:

"I think it (PND) does come from the lack of support, so if you don't have family around, you will not have the support […] I think it's because of the lack of family" (Lana).

The absence of familial support, traditionally a critical buffer against the stresses of early motherhood, seemed to exacerbate feelings of vulnerability and emotional exhaustion.

For Aisha, it went beyond assumption and into her own rooted beliefs such as "[…] having a kid in a nuclear family, in a country where you don't live anywhere near family, it's not <u>right</u>".

This longing for a 'village' during the postpartum period remained a recurrent theme which participants referred back to and was accompanied by a deepened sense of sadness and grief:

"I really did wish I had that 'village' everyone talks about […] I just wish I had more of a community around me, while that was happening (PND) 'cause that would have just made a huge difference" (Aisha).

**"It's all just on me": Bearing the weight of sole responsibility.** Through their journeys of navigating PND, participants were not only faced with the pressures of migrant mothering but alongside this had an additional sense of bearing sole responsibility.

"You have to rely on yourself. Nobody's going to look after you. There's nobody to fall back on here" (Aisha).

It may have been that by taking on sole responsibility throughout the postpartum period, participants were also attempting to cope with their own overwhelm by seeking to gain control over mothering duties: "I knew that I was anxious about everything, and I tried to control everything… and that […] was <u>consuming</u> me and it was forever present through my days and nights" (Sofi). By coping similarly in this way, Rosie recognised that it may have, in hindsight, left her feeling stuck in a vicious cycle of PND symptoms including anxiety:

"I think I was very much just trying to sort of… get control. I was trying to like read everything which I now recognise as bit of an obsessional vicious cycle that all my efforts to kind of get control has made me feel more out of control in a way."

**GET 3: "Another wave was pushing me down": Unrelenting confrontation**

This GET speaks to the participants' iterative and non-linear experiences of feeling confronted by the traumatic nature of PND, spiralling into despair and at the same time, grieving idealised expectations of motherhood. It highlights the shock that participants were faced with when experiencing PND, what remained in the aftermath. and how they continued to process the loss of what they had expected or hoped for throughout their postpartum experience.

**"Everything's survival".** Participants' personal accounts suggested that the emotional and physical impact of the postpartum period was traumatic for them. For some, like Rosie, it was the transition into becoming a mother which was experienced as an intense shock: "For a mother […] it's so physical. And so, you become so vulnerable. And <u>everything changes</u> for you so much, in a much more sort of visceral and profound way" (Rosie).

Sofi used a harrowing metaphor to highlight the unpredictability and power of the constant pressures she was faced with: "I felt like I was just in survival mode 24/7 […] It felt like… every time I would […] not even get out of the water to get some breath, but even just an <u>attempt</u> of doing that, felt like <u>another wave was pushing me down</u>"

There seemed to be a common experience of feeling powerless when faced with the relentless nature of PND. The aftermath was so intense that it eventually left some feeling paralysed: "I basically couldn't go back to work for nine months. That's how long it took me to get my head around... I couldn't even make a cup of tea without thinking, I was completely frozen in" (Valeria).

Sofi summarised her experience as follows; "for me, it feels like it was a traumatic experience […] it was the postpartum that broke me and it broke me into pieces", suggesting an enduring fragmentation of self.

**Spiralling into despair.** A vivid use of language and reference to darkness suggested PND may have tainted the whole experience of the postpartum period for participants: "There was just no like, there was nothing light about having… a baby in the UK, like it...just felt heavy and dark" (Aisha).

This overwhelming sense of despair led some participants to reflect on their increasing suicidal ideation at the time. For Monica, this seemed to be rooted in feeling completely helpless, despite her numerous attempts to seek support: "Time of

feeling, very helpless. Feeling useless. And just absolutely wanting to disappear. Because I just felt like I can't do anything right".

The hopelessness was echoed by various participants who alluded to having tried at the time to find solutions to alleviate distress, but these feeling repeatedly unsuccessful: "Nothing seems to be working" (Rosie) and "nothing worked" (Monica).

For Melany, PND was felt to have robbed her of any positive memories of her baby:

> "I have no memories [tearful] of my first child's babyhood [...] It's basically a blank. I have really no positive memories when I think about it. […] I just picture the sleeplessness and the crying and I know that there were good times, but I can't… access them".

**Grieving expectations.**  There seemed to be a common sense of disappointment transforming into enduring grief for those who looked back and struggled to recognise any positive moments: "I think there's a grief about… kind of losing those months" (Sofi). Whilst reflecting on their experiences of PND, some participants pointed out that things had not turned out how they imagined they would, "nothing got […] to work the way that I'd hoped" (Monica), suggesting a fragmentation of idealised expectations around motherhood.

Sofi recognised this in herself with hindsight, "I had this… probably <u>idealised</u> idea of how things will go. And of course, they didn't go that way" and shared the impact of falling from such expectations, which felt like a failure: "I felt like I <u>failed</u> from the beginning because I literally did not feel the joy I was <u>supposed</u> to feel, or I <u>should have</u> felt". Expectations of motherhood being joyful and exciting seemed to be a wider reflection of societal expectations towards mothers, as stated by Sofi: "Mothers are set up for failure from the very beginning".

The impact of PND not only seemed to be influenced by the individual's circumstances or events, but also by the disillusionment about motherhood that mothers were suddenly confronted with, exacerbating their sense of disappointment.

> "I wish somebody told me about becoming a mother in the UK 'cause, that's huge, that took a lot of adapting. […] It was a shock to my system, what it was like to be a mom in the UK. […] Something needs to be said about becoming a mother in the UK" (Aisha)

### GET 4: A winding journey of resilience

This GET includes participant accounts of personal and interpersonal signs of resilience throughout their journey of PND and beyond. These reflect the glimmers of hope and strength found through support groups and other forms of solidarity, primarily amongst migrant mothers. They also shed light on their personal growth along their individual healing journeys, as well as the struggles most of them continue to endure to this day.

**Finding strength in solidarity.**  Throughout their accounts of PND, there seemed to be rare but meaningful mentions of key relationships or change of circumstances which helped participants see the light at the end of the tunnel.

Monica recalled reaching out to her husband's distant relative who was from the same country of origin as her and living in the UK, referring to her as a lifeline: "She was sort of my lifeline as far as: how do I manage this?". Similarly, Aisha met a mother from the same country of origin as her, whom she intrinsically bonded with and provided her a sense of reassurance at a time when she was suffering the most:

> "<u>I can't emphasise enough</u> how important that friend was in the process of getting me out of my rut […] It was such a solid foundation, I <u>needed</u> […] that. If I had that from the beginning, I wouldn't have… I don't know if I would have needed therapy".

For Lana, the sense of solidarity was found amongst a support group she accessed after eventually seeing her GP, and this appeared to be the catalyst in feeling able to overcome her difficulties:

"I remember even just talking, listening to the other moms it felt like. Recharging and refreshing and I felt like I wasn't alone."

**A liberating shift towards healing.** On their journeys to recovery, participants noticed that although it was confronting, once they seemed able to recognise their state of mind, they were then able to start healing. For Aisha, it made the difference between feeling able to access her emotions on a deeper level and reconnect with a sense of groundedness:

"The big thing was realising that I was in survival mode. […] I had this huge realisation where I finally could feel grateful, […] I had a huge perspective shift."

This sense of gratitude also resonated by Valeria who, in hindsight, seemed to recognise that by facing PND she was also starting to overcome past traumatic experiences which had been worsening her mental health:

"Some people see it like a bad thing (PND), but I think I, after so many years, I think it's such an opportunity because if I didn't have [Child] and I didn't have my diagnosis and I didn't look into it with real eyes, I probably would be dead by now. That's the truth."

Rosie reflected on how far she had come with her husband since her experience of PND. She contemplated on how they evolved on a personal level and as a couple: "We really grew from the experience […] so there was a whole post-traumatic growth arc that we've been on", suggesting that they had managed to process some of the pain and distress experienced at the time, and reframed it into a testament of resilience.

Some participants seemed to recognise this resilience in themselves and gauged it with regards to how they felt they had coped when sharing their stories during the interviews. For Valeria, although it was "quite painful" to talk about her experience of PND, she said:

"I feel proud that I can talk about it because I wouldn't have been able to do this a year ago. And this just shows me how strong I am now and… how much… I've been through my journey. It actually feels good to, to know that. It's extremely painful. It's true. There's a lot of emotions involved."

Sofi appeared to feel proud about how far she had come in being able to recognise, accept and validate her own experience:

"I could just put a luggage down, like a bag of rocks. And it was liberating. So, I think in a way, it's liberating and it's empowering […] I feel that, I can talk about this now without breaking down into tears and… I am […] honouring this experience as it was."

**"I'm still putting myself together": The struggle continues.** The participants' testaments of resilience highlighted the complex and lengthy process of healing from and beyond PND, and although they had already come far in their journeys, there continued to be struggles.

Sofi spoke to the inherent process of healing through time after having felt 'broken' by PND, and seemed to take an active role in bringing her different parts of self, back together: "Even though the, the guilt and the shame… have been healing, there is still grief that I am navigating and processing […] I'm still putting myself together" (Sofi).

There seemed to be a common recognition that the PND diagnosis may not entirely account for the ongoing grief and broader impact of traumatic experiences that mothers continued to face beyond the first year of motherhood:

"I still have like… underlying things to work on with myself. […] It wasn't over after one year, so this kind of arbitrary cut off of like 'one year', I understand why they they do it, but really it's probably until the kid's like three… is when people need support." (Melany)

Participants often referred to the enduring psychological distress that extended well beyond the scope of their PND experiences, which they encountered as migrants in the UK:

"I think this is the major impact of… being an immigrant. I think the isolation was always gonna be there" (Valeria).

## Discussion

This study contributes to the wider literature around the intersections between migration, motherhood and mental health by offering new insight into migrant mothers' experiences of PND in the UK. It helps to partly fill the gap in research on voluntary migrant experiences [1,2]. This research was phenomenological in its focus, which meant that I was interested in understanding what the experience of *PND in migrant mothers* was like. It is likely, therefore, that it may identify experiences that are shared by migrant mothers, regardless of whether they have PND. Similarly, it may reveal experiences of distress that are shared by mothers with PND, regardless of their migration status. However, by asking participants to reflect on their experiences of PND as migrant mothers, the study positions these women in the centre of these two groups.

In this section, we present a summary of findings and discuss each of the themes in the context of wider literature.

The first GET, *Navigating layers of disconnect*, contributes to research highlighting assimilation and cultural identity as central to the relationship between migration and mental health distress [41]. It suggests links between migration, motherhood and PND through multiple disconnections with self and others.

The subtheme *"It doesn't feel like my culture": Cultural displacement* evokes the impact of migration on participants' cultural identities and acculturation, defined as "changes that take place as a result of contact with culturally dissimilar people, groups and social influences" [42]. Acculturation involves loss of culture and support systems alongside adaptation to new environments [43]. Participants evoked this sense of cultural confusion and feelings of alienation, which are common in the literature about post-migration stressors [41,44]. Roitman highlights the uniqueness of migrant mothers renegotiating their national identity and new role as mothers simultaneously [45].

Research suggests there is a higher risk for migrants to experience a sense of alienation due to the complex and multiple adjustments they face when adapting to a new environment [46]. This seemed to contribute to participants' sense of isolation, which is deemed a risk factor for mental health disorders [44]. Without social support, isolation can lead to a sense of rejection and low self-esteem [44]. This was echoed within participants' experiences of feeling left out and misunderstood, withing the subtheme *Grappling with old and new relationships*.

Research suggests that lack of social support and poor interpersonal relationships are risk factors for PND [47]. Participants described a sense of alienation even within their own partnerships. This was reflected in the subtheme *Feeling estranged*, which highlighted marital tensions postpartum and exacerbated loneliness. Alongside difficulties in relationships with others, participants also struggled in their relationship to self. Through migration and PND, participants experienced a multi-layered shift in identity and sense of self. Social identity theory explains self-concept as derived from group membership [48–50], providing context for the subtheme *"Always around but not present": Loss of self*. Participants

seemed to have lost a sense of group membership by virtue of being migrants which could partly explain their sense of losing themselves. The experiences of feeling "alien to their core, internal or authentic selves" [51] also reflect the fragmentation of self which is central to PND [44,52].

The second GET, *The burden of "learning the ropes by yourself"*, highlights layers of learning faced alone through navigating a new country while becoming a mother. In *Building everything from nothing*, participants described the dual shock of the new: becoming mothers and doing so in a new country, leaving them in unfamiliar territory, with additional pressures of economic and immigration status [53]. Research suggests that migrant women are more vulnerable to mental health difficulties during the postpartum period compared to native women, due to lack of access to support and local resources, which supports the finding that they are building a new family life with limited existing resources [54].

An *isolating silence* left participants struggling alone with PND. A UK meta-synthesis identified stigma as a main factor influencing women's decision to seek help for perinatal distress [55]. While stigma was not explicitly mentioned in this study, it may have shaped the silence participants described experiencing. This reflects Tobin et al.'s [56] findings of mothers "suffering in silence (and solitude) from an invisible illness".

Through facing new motherhood alone and experiencing layers of silence around their journeys of PND, participants shared a sense of *Longing for a 'village'*. Roitman [45] similarly found migrant mothers reported homesickness only after giving birth. In our study, participants related their experience of PND to the physical distance from families and friendships, echoing Schmied et al.'s [57] findings on increased stress linked to lack of community in host countries. Participants' lamentation regarding lack of community was also reflected in Foster's [58] study of migrant mothers.

The weight of responsibility was captured in *"It's all just on me": Bearing the weight of sole responsibility*. Mothers in general can experience a sense of overwhelm related to their mothering responsibilities when suffering from PND [53,59]. Homewood et al. [59] describe an 'overwhelming responsibility phase' in motherhood, where responsibilities feel unmanageable. Participants in this study similarly doubted maternal adequacy, heightening guilt, shame, and fear, which perpetuated PND.

Ettinger De Cuba et al. [60] argue that migrant mothers not only face the mental load of raising children and managing households with limited resources, but that this is compounded by wider political and social contexts. They highlight "structural vulnerability to describe economic and political processes that impose physical and emotional suffering […] in a structured manner" [60], shifting focus from individual vulnerability to broader structures.

The third GET, *Unrelenting confrontation: "Another wave was pushing me down"*, reflects the relentlessness and hopelessness of PND, and how participants felt these feelings tainted their postpartum period. In *"Everything's survival"*, participants described PND as survival, resonating with Beck's [61] four-stage theory of PND. Stage 3, 'struggling to survive', captures survival struggles particularly acute for migrant mothers [61]. Participants appeared in fight-or-flight states postpartum, with physical symptoms entwined with survival, consistent with descriptions of PND as debilitating [57].

Hopelessness and helplessness were central to *"Spiralling into despair"*, echoing Beck's [62] conceptualisation of PND through the theme 'spiralling downwards'. Hopelessness is a characteristic of PND included in the Edinburgh Postnatal Depression Scale [63]. Women with PND are at increased risk of suicidal behaviour, particularly within the first year of diagnosis [64]. These risks may be heightened for migrant mothers with fewer support options.

In *Grieving expectations*, participants not only described the "shock of the new" [53], but also of the discrepancy between expectations and reality of migrant motherhood [44]. Unrealistic ideals of motherhood can exacerbate distress when women do not experience expected fulfilment [59]. Discrepancies between expectations and reality are linked to PND [59,65]. Feminist literature argues some women's reports of PND may reflect typical adjustment difficulties amplified by unmet expectations [59]. Participants grieved idealised expectations and opportunities, consistent with Mollard's [66] theme 'crushed maternal role expectation' and Beck's [67] identification of incongruity between expectations and reality as central to PND.

The final GET, *A winding journey of resilience*, encompassed narratives of growth. This contrasts with dominant literature emphasising vulnerabilities of migrants [68,69] and mothers with PND [70,71]. Participants described resilience through solidarity, recognising PND, seeking support, and sharing experiences.

*Finding strength through solidarity* captured the role of forming relational connections in validating PND experiences. Although difficult to build, such bonds enable further help seeking behaviours [62]. Participants described peer support as key to their recovery. This aligns with Cronin's [72] findings on friendships in motherhood and Benchekroun's [73] work on solidarity among migrant mothers. The emotional support from these connections were simply conceptualised as "life lines".

*A liberating shift towards healing* highlighted participant reflections on the importance of recognition and validation of PND, which supported their growth. For some, professional recognition "opened the doors" gaining a better understanding and accessing support. As in Beck's [62] synthesis, recognition allowed participants to regain control. Sharing experiences of PND helped reclaim selfhood and foster recovery, with some expressing pride in participation, similar to Stone and Kokanovic's [74] findings. Williams [75] also identified recovery as empowerment.

Despite resilience, participants acknowledged ongoing struggles. *"I'm still putting myself together": The struggle continues* captured ongoing challenges participants faced beyond PND. Beck [76] noted mothers may grieve postpartum moments lost to depression. Participants in the current study described recovery as ongoing, still struggling with isolation and lack of childcare even years later. This reflects findings from studies of migrant mothers facing long-term difficulties [77–79].

## Implications

The findings of this study indicated that migrant mothers felt a burdening silence and sense of isolation within their experiences of PND. This seemed to fuel their interest in helping to raise awareness by participating in the study, with the hope that this would reach other migrant mothers and help them feel less alone in their experiences of PND. By foregrounding and disseminating migrant mother stories, this study contributes to existing research which highlights the importance of tackling the sense of isolation participants voiced through raising awareness [80,81].

It is important for awareness around common experiences of isolation and other migration challenges that may impact mental health to be raised amongst various migrant communities, from a community-led perspective. Local community initiatives – such as having NHS or charity staff hold a stall at community or local events with the aim to raise awareness around postnatal distress – are necessary, not only to promote migrant mothers' sense of belonging but also to generate a greater sense of community wellbeing through fostering social connections and inclusivity [82,83].

For migrant mothers who do manage to access NHS services within the first year of giving birth, it is particularly important for staff to have a greater understanding of the potential layers of risks and challenges they face which may be contributing to their overall presentation. This awareness could prompt further opportunities for screening for PND and for conducting tailored assessments which include particular attention to their experience of being migrant mothers. Building on Wittkowski et al.'s [23] recommendations, healthcare professionals should be supported, for example through training, with cultural awareness and culturally appropriate practices when working with migrant mothers.

The wider impact of migrant mothers' experiencing PND on their partners, babies and family was also acknowledged, suggesting that support could be offered from a systemic lens, whether or not that directly includes others depending on individual circumstances [84,85].

## Limitations and future research

Recruitment was carried out via social media using mixed purposive and snowballing techniques. This created a limitation whereby recruitment excluded certain participants such as those who are not able to or do not access social media. This approach inadvertently created an economic privilege and ableist gap between voices which were most and least likely to

be heard. Future research should include in person recruitment strategies such as joining community forums or sharing the poster in cultural hubs.

Participants in our study had a fluent level of English, which excluded migrant mothers who may not feel confident with their English, and/or who felt less able to talk about emotional experiences in English. Future research should explore the use of interpretation services in order to reach a wider pool of participants.

There was a majority of White women who participated in our study. Further research could focus particularly on the voices of Black and Brown migrant mothers and their experiences of postnatal distress, which are currently significantly underrepresented [86–88].

Existing literature identifies partner involvement as crucial in addressing mothers' distress during the postnatal period [89,90]. In our study, we opted to focus solely on mothers' experiences. Building on Atkinson et al.'s [91] qualitative evidence synthesis and Vo et al.'s [92] systematic review, further research can explore migrant fathers' experiences of the postpartum period.

## Conclusion

Overall, this research showed that migrant mothers' experiences of PND in the UK are layered, complex and unique due to the intersecting experiences of becoming mothers while also being migrants. This dual identity, compounded by cultural barriers and limited access to support amongst other factors, seemed to contribute to the challenges faced during and beyond experiences of PND. The findings contribute to the breadth of existing literature around experiences of PND, whilst highlighting the additional complexities influenced by a context of voluntary migration.

## Supporting information

**S1 File. Project timeline.**
(DOCX)

**S2 File. Screening call guide.**
(DOCX)

**S3 File. Interview guide.**
(DOCX)

## Author contributions

**Conceptualization:** Laetitia Rater.

**Data curation:** Laetitia Rater.

**Formal analysis:** Laetitia Rater.

**Investigation:** Laetitia Rater.

**Methodology:** Laetitia Rater.

**Project administration:** Laetitia Rater.

**Resources:** Laetitia Rater.

**Supervision:** Debra Marais, Aisling Kelly.

**Validation:** Laetitia Rater.

**Visualization:** Laetitia Rater, Debra Marais.

**Writing – original draft:** Laetitia Rater.

**Writing – review & editing:** Laetitia Rater, Debra Marais, Aisling Kelly.

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
