## [Decision Letter · Decision Letter 0]

5 Feb 2026

Dear Dr. Rater,

Thank you for submitting your manuscript to PLOS ONE. After careful consideration, we feel that it has merit but does not fully meet PLOS ONE’s publication criteria as it currently stands. Therefore, we invite you to submit a revised version of the manuscript that addresses the points raised during the review process.

Reviewer #1: Introduction- Well-researched and evidence-based.Provides important context on migration and perinatal mental health.Clear identification of a meaningful research aim.

Methodology: appropriately positioned for exploring the deeply personal, identity-based and culturally situated experiences of migrant mothers with postnatal depression (PND). Well written, ethically robust, and grounded in appropriate qualitative and phenomenological principles.

Analysis- The narrative is compelling, the themes are coherent, and the use of participant quotations is strong.

Limitations The analysis does not sufficiently acknowledge how the researcher’s background, assumptions or interpretative lens shaped the analytic process. A reflexivity section is recommended, detailing:

researcher positionality

emotional and cognitive responses during analysis

potential biases

steps taken to mitigate interpretative influence

Discussion, conclusion, limitations- Deep, thoughtful integration of participants’ experiences with theory.

Transparent limitations and clear recommendations for future research

Reviewer #2: Good topic.

If possible try to take more samples for reliable results.Also addition of Gantt chart is advisable for qualitative studies.

Minor corrections required in terms of samle size calculation and minimum sample size numbers.

Reviewer #3: The manuscript is written well, with a very good explanation of IPA. However, it lacks specificity as mentioned in the attached file. Data and tools are not available (as off now) to understand the data collection flow and validity. The authors could revise the manuscript as suggested and provide the access to study tool (interview guide) and the qualitative data for a better understanding.

We look forward to receiving your revised manuscript.

Kind regards,

Pracheth Raghuveer, MD, DNB

Academic Editor

PLOS One

Journal Requirements:

4. Please include captions for your Supporting Information files at the end of your manuscript, and update any in-text citations to match accordingly. Please see our Supporting Information guidelines for more information: http://journals.plos.org/plosone/s/supporting-information....

5. We note that there is identifying data in the Supporting Information file < aLMS PGR UH 05737(1) Rater L 19000424 Notification (1) and aLMS PGR UH 05737 Rater L 19000424 Notification>. Due to the inclusion of these potentially identifying data, we have removed this file from your file inventory. Prior to sharing human research participant data, authors should consult with an ethics committee to ensure data are shared in accordance with participant consent and all applicable local laws.

-Location data

Please remove or anonymize all personal information (Name), ensure that the data shared are in accordance with participant consent, and re-upload a fully anonymized data set. Please note that spreadsheet columns with personal information must be removed and not hidden as all hidden columns will appear in the published file.

Reviewers' comments:

Reviewer's Responses to Questions

**Comments to the Author**

1. Is the manuscript technically sound, and do the data support the conclusions?

Reviewer #1: Yes

Reviewer #2: Yes

Reviewer #3: Partly

2. Has the statistical analysis been performed appropriately and rigorously?

Reviewer #1: N/A

Reviewer #2: Yes

Reviewer #3: N/A

3. Have the authors made all data underlying the findings in their manuscript fully available?

Reviewer #1: Yes

Reviewer #2: Yes

Reviewer #3: No

4. Is the manuscript presented in an intelligible fashion and written in standard English?

Reviewer #1: Yes

Reviewer #2: Yes

Reviewer #3: Yes

Reviewer #1: Introduction- Well-researched and evidence-based.Provides important context on migration and perinatal mental health.Clear identification of a meaningful research aim.

Methodology: appropriately positioned for exploring the deeply personal, identity-based and culturally situated experiences of migrant mothers with postnatal depression (PND). Well written, ethically robust, and grounded in appropriate qualitative and phenomenological principles.

Analysis- The narrative is compelling, the themes are coherent, and the use of participant quotations is strong.

Limitations The analysis does not sufficiently acknowledge how the researcher’s background, assumptions or interpretative lens shaped the analytic process. A reflexivity section is recommended, detailing:

researcher positionality

emotional and cognitive responses during analysis

potential biases

steps taken to mitigate interpretative influence

Discussion, conclusion, limitations- Deep, thoughtful integration of participants’ experiences with theory.

Transparent limitations and clear recommendations for future research

Reviewer #2: Good topic.

If possible try to take more samples for reliable results.Also addition of Gantt chart is advisable for qualitative studies.

Minor corrections required in terms of samle size calculation and minimum sample size numbers.

Reviewer #3: The manuscript is written well, with a very good explanation of IPA. However, it lacks specificity as mentioned in the attached file. Data and tools are not available (as off now) to understand the data collection flow and validity. The authors could revise the manuscript as suggested and provide the access to study tool (interview guide) and the qualitative data for a better understanding.

.

Reviewer #1: No

Reviewer #2: **Yes:** Algotar Priyank DineshkumarAlgotar Priyank DineshkumarAlgotar Priyank DineshkumarAlgotar Priyank Dineshkumar

Reviewer #3: **Yes:** Sandipta ChakrabortySandipta ChakrabortySandipta ChakrabortySandipta Chakraborty

---

## [Author Response · Author response to Decision Letter 1]

8 Mar 2026

We appreciate the time that was taken to review our paper and the thoughtful feedback that has been provided. Please find the specific reviewer and editor comments responses addressed and outlined in tables within the 'Response to reviewers' and 'Additional requirements' documents.

---

## [Decision Letter · Decision Letter 1]

5 Apr 2026

Migrant mothers' experiences of postnatal depression in the UK

PONE-D-25-47137R1

Dear Dr. Rater,

We’re pleased to inform you that your manuscript has been judged scientifically suitable for publication and will be formally accepted for publication once it meets all outstanding technical requirements.

Kind regards,

Pracheth Raghuveer, MD, DNB

Academic Editor

PLOS One

Additional Editor Comments (optional):

Reviewers' comments:

Reviewer's Responses to Questions

**Comments to the Author**

Reviewer #2: All comments have been addressed

Reviewer #3: All comments have been addressed

2. Is the manuscript technically sound, and do the data support the conclusions?

Reviewer #2: Yes

Reviewer #3: Yes

3. Has the statistical analysis been performed appropriately and rigorously?

Reviewer #2: N/A

Reviewer #3: N/A

4. Have the authors made all data underlying the findings in their manuscript fully available?

Reviewer #2: Yes

Reviewer #3: No

5. Is the manuscript presented in an intelligible fashion and written in standard English?

Reviewer #2: Yes

Reviewer #3: Yes

Reviewer #2: The manuscript addresses an important and underexplored topic—experiences of postnatal depression (PND) among voluntary migrant mothers in the UK—using an appropriate qualitative approach. The use of Interpretative Phenomenological Analysis is well-justified and aligns with the study’s aim of capturing lived experiences. The themes are rich, coherent, and supported by illustrative participant quotes, enhancing credibility and depth.

However, several areas require improvement. First, the sample size (n=7), while acceptable for IPA, limits transferability; the authors should better justify sample adequacy and discuss saturation more explicitly. Second, there is limited diversity in the sample (predominantly White participants), which restricts representation of more vulnerable migrant groups—this should be more critically reflected upon. Third, while reflexivity is well-described, the potential influence of researcher positionality on theme development could be more explicitly linked to findings.

The methodology section would benefit from clearer details on recruitment (e.g., response rate) and how rigor was ensured (e.g., triangulation, audit trail). Additionally, the data availability statement does not fully align with journal requirements and may need revision.

Overall, the study is meaningful and publishable.

Reviewer #3: All comments have been addressed. Regarding data availability, authors mentioned that disclosing the anonymous qualitative transcripts is not possible due to potential ethical violation, which is acceptable.

However, inserting page numbers in the submitted document would have been better to track the modifications made more easily.

.

Reviewer #2: **Yes:** Algotar Priyank DineshkumarAlgotar Priyank DineshkumarAlgotar Priyank DineshkumarAlgotar Priyank Dineshkumar

Reviewer #3: **Yes:** Sandipta ChakrabortySandipta ChakrabortySandipta ChakrabortySandipta Chakraborty

---

## [Editor Report · Acceptance letter]

PONE-D-25-47137R1

PLOS One

Dear Dr. Rater,

I'm pleased to inform you that your manuscript has been deemed suitable for publication in PLOS One. Congratulations! Your manuscript is now being handed over to our production team.

Kind regards,

on behalf of

Dr. Pracheth Raghuveer

Academic Editor

PLOS One